# Intelligent Task Offloading in Fog Computing Based Vehicular Networks

**Ahmad Naseem Alvi** [1], **Muhammad Awais Javed** [1], **Mozaherul Hoque Abul Hasanat** [2,*],
**Muhammad Badruddin Khan** [2], **Abdul Khader Jilani Saudagar** [2], **Mohammed Alkhathami** [2] and **Umar Farooq** [1]

1   Department of Electrical and Computer Engineering, COMSATS University, Islamabad 45550, Pakistan;
    naseem_alvi@comsats.edu.pk (A.N.A.); awais.javed@comsats.edu.pk (M.A.J.); umar4420@gmail.com (U.F.)
2   Information Systems Department, College of Computer and Information Sciences, Imam Mohammad Ibn
    Saud Islamic University (IMSIU), Riyadh 11432, Saudi Arabia; mbkhan@imamu.edu.sa (M.B.K.);
    aksaudagar@imamu.edu.sa (A.K.J.S.); maalkhathami@imamu.edu.sa (M.A.)
*   Correspondence: mhhasanat@imamu.edu.sa

**Abstract:** Connected vehicles in vehicular networks will lead to a smart and autonomous transportation system. These vehicles have a large number of applications that require wireless connectivity by using cellular vehicle-to-everything (C-V2X). The infrastructure of C-V2X comprises multiple roadside units (RSUs) that provide direct connectivity with the on-road vehicles. Vehicular traffic applications are mainly categorized into three major groups such as emergency response traffic, traffic management and infotainment traffic. Vehicles have limited processing capabilities and are unable to process all tasks simultaneously. To process these offloaded tasks in a short time, fog servers are placed near the RSUs. However, it is sometimes not possible for the fog computing server to process all offloaded tasks. In this work, a utility function for the RSU to process these offloaded tasks is designed. In addition, a knapsack-based task scheduling algorithm is proposed to optimally process the offloaded tasks. The results show that the proposed scheme helps fog nodes to optimally scrutinize the high-priority offloaded tasks for task execution resulting in more than 98% of emergency tasks beingprocessed by fog computing nodes.

**Keywords:** vehicular networks; fog computing; task offloading

## 1. Introduction

The vehicular network is a vital component of future smart cities. Smart connectivity in vehicular networks helps in traffic management, increased safety, and better infotainment systems [1–8]. Efficient traffic management reduces traffic congestion, resulting in reduced carbon emissions thus improving atmospheric conditions in future smart cities with reduced accidents [9–19]. In addition, route guidance applications, predictive maintenance of vehicles and infotainment services will benefit from vehicular networks [20–24].

Vehicular networks are comprised of vehicle's built-in on-board units (OBU), geographical placement of roadside units (RSU) and city traffic command centres (TCC) [18,25]. Vehicles communicate through their wireless transceivers called OBUs, which are responsible for transmitting and receiving all types of vehicle information. In vehicle to vehicle (V2V), vehicles communicate with each other through their OBUs. RSUs are geographically located across roads in such a way that vehicles during their journey remain connected with one of the RSU located on the road. These RSUs are backward connected with the TCC through the Internet cloud for collecting and analyzing the traffic data and also providing infotainment services to the vehicles.

With the rapid growth of 5G wireless technology, the concept of autonomous vehicles has become a reality [26–31]. One of the key features of the 5G cellular vehicle-to-everything (C-V2X) standard is the reduced latency needed for the autonomous driving application.

However, to cope up with these latency requirements, C-V2X has to propose some modifications in its medium access control and physical layers.

Vehicular communication relies on wireless technologies such as cellular communications, thus leading to intelligent transportation systems (ITS). ITS offers multiple smart services such as unmanned autonomous vehicle driving, online gaming, augmented and virtual reality. Most of these services require high computational capability and do not compromise delay. Moreover, an increase in high traffic applications along with simultaneous multiple tasking is expected from vehicles in the near future. To meet these challenges, the vehicles require high computational and battery sources. Though offloading high computational tasks and data traffic through the Internet cloud solves these challenges, computing at remote locations through centralized cloud-based Internet access may cause significant delays that are not acceptable in delay-sensitive applications of ITS.

To mitigate these delay-sensitive issues, computing servers are deployed on each RSU location and are connected with each RSU to behave as a fog node. These fog nodes have abundant caching and computational capability to perform vehicles tasks with cache and computing capabilities [32–35]. However, the fast speed with a limited coverage area provides shorter connectivity time for vehicles with RSUs and is a major constraint in task computation. Moreover, due to a large number of different varieties of tasks, a fog node cannot execute all tasks simultaneously. This causes a delay in the execution of some tasks that are not required for different applications and may badly affect the performance of the vehicular networks.

In this work, an efficient task execution by fog node ($ETE_{FN}$) that gives task preference in accordance with time constraints is proposed . An algorithm along with a superframe structure is designed to fulfill these requirements. The salient features of the $ETE_{FN}$ are given below:

1. Screen out those offloaded tasks that are about to leave the RSU coverage area. The screen-out tasks are forwarded to a cloud server for task execution.
2. A utility function is designed for the rest of the offloaded task requests according to task preferences and the remaining time of attachment with the RSU.
3. If task requests are more than the execution capacity of the fog node, tasks are optimally scrutinized according to their utility function by applying the 0/1 knapsack algorithm. The scrutinized tasks are executed at the fog node, and the rest of the tasks are forwarded to the cloud for processing.

The rest of the paper is organized as follows: Section 2 discusses fog computing and different research about task offloading schemes. The system model is discussed in Section 3 The proposed task execution policy is examined in Section 4. The performance of the proposed scheme is evaluated in Section 5, and Section 6 concludes the paper.

## 2. Related Works

The concept of fog computing is preferred over cloud computing due to nearby processing with reduced delay. That is why there are multiple research studies on the prospects of fog computing. In vehicular networks, the placement of a server adjacent to a RSU creates a fog node. The utility of fog computing is being heavily studied due to its high utilities in vehicular networks such as content caching and task processing.

An intelligent technique for task offloading to edge nodes by using a finite horizon Markov decision has been proposed for vehicular networks [36]. The main focus of this work determines the transition probabilities to offload the task to that edge node by considering driver behaviour, communication behaviour and road topology. The authors claim that their proposed task offloading scheme reduces delay significantly.

In [37], a double bipartite matching task offloading algorithm is proposed for high-speed vehicles that are categorized in three different states. In this work, vehicles offload tasks to their nearby edge node, and the edge node can further forward the offloaded task to the next edge node that is in the direction of vehicle movement. However, the authors have not considered the task processing capacity of the next edge node.

The authors in [38] proposed a task offloading scheme by optimizing the fog node selection. The authors execute the offloaded tasks through a load balancing technique by introducing fiber wireless technology in which all RSUs are backwardly connected through fiber with a software-defined network for information management and a centralized network. The software-defined networking node forwards the offloaded tasks to the edge node by applying the game theory. The authors in this work reduce the delay by applying offloading techniques that includes complex and costly fiber wireless technology. Moreover, the offloaded tasks are routed to a software-defined network through roadside units, causing unnecessary delay.

Chao et al., in [39] proposed a mobility aware task offloading scheme by considering the maximum allowed latency of an offloaded task along with the communication and computing capabilities of allocated fog node resources. The authors proposed a greedy scheme to assign tasks to other nearby vehicles in addition to a bipartite matching algorithm and claimed that their proposed scheme reduces the execution time of offloaded tasks. In [40], an adaptive learning task offloading algorithm was proposed for offloading task execution of vehicles. The algorithm is based on the multi armed bandit theory to minimize the offloaded task delay.

All the abovementioned task offloading schemes for vehicular networks are either based on vehicle to vehicle bases or offer different load balancing schemes on different criteria basis. However, no one has proposed an efficient task offloading scheme for fog node, and no one has discussed the processing capacity of fog computing nodes. In this work, an optimal task offloading scheme is proposed by considering the task processing capacity of fog computing nodes.

## 3. System Model

The system model used in this paper is shown in Figure 1. All vehicles are connected with RSUs by using V2I communication. These RSUs are placed on the roadside at different geographical areas to provide the coverage area of the complete highway. These RSUs are attached with servers for efficient computing to make them fog-computing nodes. These fog nodes are backwardly connected with the Internet cloud. Vehicles move with uniform velocity, and the expected time of attachment with each RSU is the same. Vehicles have one or more different types of tasks that are required to be executed. These tasks are mainly divided into three different types, such as emergency or safety tasks, traffic management tasks and infotainment tasks. The time sensitivity of execution of each type of task varies along with their task size. Vehicles want to offload these tasks for timely execution.

In this work, we consider $R$ RSUs, and each RSU is connected with $V$ vehicles on the road. Each vehicle $n_i$ attached with a RSU has one to three tasks for execution. These tasks are categorized as $t_1$, $t_2$ and $t_3$ and are prioritized from low to high priority level, respectively. $T$ is the total number of tasks that are required to be executed and are calculated as:

$$T = \sum_{i=1}^{V} \sum_{j=1}^{k} n_i(t_j) \tag{1}$$

The downloading data rate for a vehicle to download a task solution from a fog node ($DR_1$) is computed as:

$$DR_1 = log_2(1 + \gamma_{v,f}) \tag{2}$$

where $\gamma_{v,f}$ is the signal to noise ratio between the vehicle and the fog computing node.

The data rate for downloading executed tasks from a cloud node such as TCCC ($DR_2$) is calculated as:

$$DR_2 = log_2(1 + \gamma_{v,c}) \tag{3}$$

where $\gamma_{v,c}$ is the signal to noise ratio between a vehicle and a cloud node. We assume that there are multiple channels available in each fog node to transmit all executed tasks simultaneously without considering any queuing delay. We used the path loss propagation model along with Nakagami-m multipath fading [41] to find the RSU coverage range.

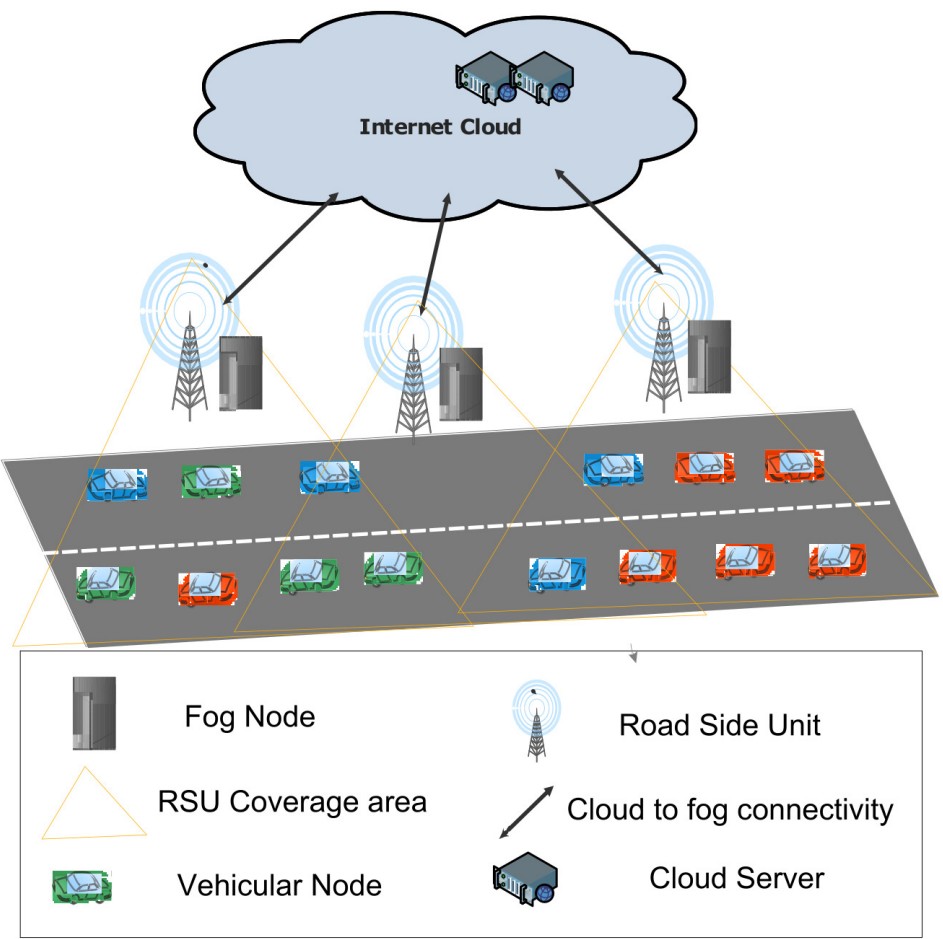

**Figure 1.** System model.

## 4. Proposed Task Execution Policy

This work proposes an efficient task execution policy for fog nodes ($ETE_{FN}$) that executes those tasks that are offloaded by vehicles during their journey in vehicular networks. Fog nodes located near RSUs execute these offloaded tasks to minimize the task delay. A fog node has limited processing capability compared to cloud servers, and tasks beyond its capacity cannot be processed. In addition, there are different categories of tasks with varying delay constraints. It is therefore required that the fog node optimally scrutinize these offloaded tasks to execute in accordance with their task emergency. $ETE_{FN}$ optimally scrutinizes offloaded tasks according to their sizes and priorities in the following way:

- A task selection policy is introduced by excluding offloaded tasks of those vehicles that are about to leave the fog node coverage area.
- A utility function of the fog node determines the priority for all offloaded tasks by all vehicles in the range of the RSU.
- An optimal selection of offloaded tasks to be executed by the fog node are determined by applying a 0/1 knapsack algorithm.

### 4.1. Task Selection Policy

A fog node in $ETE_{FN}$ after regular intervals calculates the total number of received tasks and makes the decision whether to process these tasks on its own or forward them to the cloud server based on the following conditions:

1. If the vehicle's remaining time of attachment with the RSU is less than its executed task downloaded time, then the task is forwarded to the cloud for task execution, and the cloud is supposed to download the task to a fog node placed at the vehicle's next attached RSU.

2. If all the valid requested tasks are less than its task execution capacity, then it processes all tasks itself, and no task will be forwarded to the cloud server.

A complete task processing criteria is shown in Algorithm 1.

---

**Algorithm 1:** Task processing Criteria

---

1  $y_i \leftarrow$ *Current task*
2  $t_i \leftarrow$ *Remaining time of attachment with RSU*
3  $T_c \leftarrow$ *Executed task downloading time*
4  $i \leftarrow$ *current vehicle ID*
5  $C \leftarrow$ *Task execution capacity of fog node*
6  $Y \leftarrow$ *Max. no. of tasks requested by a vehicle*
7  $V \leftarrow$ *Max. no. of tasks requesting vehicles*
8  $y_i \leftarrow$ *content size requested by $i^{th}$ content*
9  **Task execution policy**

---

10 **for** *x = 1 to v* **do**
11      **for** $y_x = 0$ *to Y* **do**
12          **If** $t_i \leq T_c$    $\leftarrow$ *Forward the task to cloud server*
13          **Else**
14          $\leftarrow$ *Include the task in task processing list x = x + 1*
15          **EndIf**
16      **end**
17 **end**
18 $\leftarrow$ *Compute all tasks (K) in task processing list*
19 **If** $K \leq C$    $\leftarrow$ *Fog node executes all tasks*
20 **Else**
21 $\leftarrow$ *Fog node apply 0/1 knapsack to scrutinize tasks*
22 **EndIf**

---

If requested tasks are more than the task execution capacity, then it optimally scrutinizes tasks to be performed at the fog node by applying 0/1 knapsack, and the rest of the tasks are forwarded to the cloud server. The value of 0/1 knapsack in the proposed scheme is derived from a utility function that is computed by the fog node for each offloaded task.

*4.2. Fog Node Utility Function*

The utility function is calculated to determine the value of each vehicle's offloaded task. The utility function is based on the task priority and the remaining time of attachment of the vehicle with the requested fog node. The higher the priority of the tasks, the higher its utility value will be. Similarly, the smaller the time of attachment of the vehicle with the requested fog node, the higher its value will be. In case two vehicles offload the same priority tasks to the fog node, then the fog node prefers to execute those tasks that are going to leave this fog node earlier first.

The fog node computes a utility function for each task offloaded by vehicles by considering the task sensitivity and the vehicle's remaining time of attachment with that fog node. Suppose there are two emergency tasks; then, priority should be given to the one that is about to leave the RSU.

For the *i*th vehicle offloaded task, the fog node calculates its utility function according to its task priority ($\Omega_i$) and its remaining time of contact ($T_{rem(i,f)}$) with the fog node. The utility function of ith vehicle ($U_i$) is calculated as:

$$U_i = \frac{\Omega_i}{T_{rem(i,f)}} \tag{4}$$

where $T_{rem(i,f)}$ is calculated as:

$$T_{rem(i,f)} = \frac{d_{i,f}}{V_i} \tag{5}$$

where $d_{i,f}$ is the remaining distance of $i$th vehicles before leaving the fog node that can be determined from GPS, and $V_i$ is the speed of the vehicle.

The fog node in $ETE_{FN}$, after computing the utility function, applies, the 0/1 knapsack algorithm to optimally scrutinize the offloaded tasks for execution.

*4.3. 0/1 Knapsack for Task Scheduling*

Fog nodes have limited execution capacity, and if requested tasks are more than their capacity then they need to scrutinize delay-sensitive tasks.

Optimal scrutiny of different-sized task requests in the fixed data execution capacity of the fog node is solved by the 0/1 knapsack algorithm. The decision is based on the utility function of each offloaded task request as discussed in Section 4.2.

Suppose the task execution capacity of a fog node is $C$ data/cycle, and there are different sizes of $K$ task requests by $V$ vehicular nodes. These $K$ tasks are categorized into three types of tasks as infotainment, traffic management and emergency tasks and are prioritized from low, medium and high, respectively. If each of $V$ vehicle attached with the RSU, sends one to three task requests to the fog node, then the knapsack optimization technique is applied with the following constraint:

- The requested offloaded tasks of $V$ vehicles, with $t$ requested tasks by each vehicle should be less than the task capacity $C$ and is represented as:

$$\sum_{i=1}^{V} \sum_{j=1}^{k} n_i(t_j) \leq C$$

- The scrutinized tasks are selected with maximum value such as utility.

$$Max \sum_{j=1}^{K} U_j d$$

The knapsack optimizes the scrutiny process by filling a knapsack table, and from this table, an optimal task with higher utility values is selected. A complete knapsack algorithm is shown in Algorithm 2.

---

**Algorithm 2:** Task Selection Criteria

---

1   $y \leftarrow$ *Current task size*

2   $Y \leftarrow$ *Max. task execution capacity of fog node*

3   $i \leftarrow$ *Task ID*

4   $n \leftarrow$ *Max. no. of tasks*

5   $X[i, y] \leftarrow$ *Cell value of table with $i^{th}$ task and y processing*

6   $y_i \leftarrow$ *task size requested by $i^{th}$ task*

7   **filling of knapsack table:**

8   **for** $y = 0$ *to* $Y$ **do**

9      $X[0, y] = 0$

10      // Initialize 1st row to 0's

11   **end**

12   **for** $i = 1$ *to* $v$ **do**

13      $X[i, 0] = 0$

14      // Initialize 1st column to 0's

15   **end**

16   **for** $i = 1$ *to* $v$ **do**

17      **for** $y = 0$ *to* $Y$ **do**

18          **If** $y_i \leq y$ **If** $y_i + X[i-1, y-y_i] > X[i-1, y]$

19          $X[i, y] = y_i + X[i-1, y-y_i]$

20          **Else**

21          $X[i, y] = X[i-1, y]$

22          **EndIf**

23          **Else**

24          $X[i, y] = X[i-1, y]$

25          **EndIf**

26      **end**

27      Initialize i and w:

28      $v \leftarrow i$

29      $Y \leftarrow m$

30   **end**

31   **optimal task selection:**

32   **while** $i > 1$ *and* $y > 1$ **do**

33      **If** $B[i, y] > B[i-1, y]$

34      $i^{th}$ content is included in optimal solution

35      $i = i - 1$

36      $y = y - y_i$

37      **Else**

38      $i = i - 1$

39      **EndIf**

40   **end**

---

## 5. Performance Evaluation

In this section, the performance of $ETE_{FN}$ is evaluated in different prospects. The offloaded tasks are categorized in three levels of priority. Each vehicle attached with the fog node is randomly chosen to offload one to three tasks of different levels to its fog node as discussed in Section 3. The performance of $ETE_{FN}$ is evaluated in different scenarios, and it is compared with the following three task offloading schemes:

1. Tasks of different vehicles are offloaded to a fog node for task computation. Offloaded tasks are processed by following the smallest task first (STF) algorithm. This allows the fog node to start executing tasks from the smallest tasks and keeps on executing

the tasks to the task processing capacity of the fog node. The remaining tasks are forwarded to the cloud for computation and execution.

2. In the second scheme, a fog node executes tasks by following the longest task first (LTF) algorithm. Contrary to STF, LTF allows a fog node to start executing from the longest task execution and keeps on processing to the task processing capacity of the fog node. The remaining tasks are forwarded to the cloud for computation and execution.

3. The fog node processes offloaded tasks up to its processing capacity by applying first come first serve (FCFS) mechanism, and the rest of the tasks are forwarded to the cloud for processing and execution.

We developed a simulation environment to evaluate the performance of the proposed technique with two other schemes. The detailed simulation setup along with its different parameter values are described in Section 5.1 and results are discussed in Section 5.2.

### 5.1. Simulation Parameters and Performance Metrics

The performance of the proposed work was evaluated using simulations conducted in MATLAB. The model used for simulations was discussed in Section 3. In this simulation, task offloading with different task types was performed. These task sizes are ranged from 5 kB to 20 kB for sensitive tasks, 12 kB to 30 kB for traffic management tasks, and 20 kB to 50 kB for infotainment tasks. The coverage area of a fog node was taken as 200 m.

The data rate for the downlink between the fog node and the vehicle is 8 Mbps. However, the downloading data rate of a vehicle from a cloud server is 2 Mbps. A fog computing node has limited processing capability and can execute 500 kBytes of tasks simultaneously, However, the cloud has unlimited processing capacity.

A list of salient simulation parameters are shown in Table 1. In this work, a Monte-Carlo-based simulation was performed for fair comparative analysis of $ETE_{FN}$. The results are obtained as an average of $10^3$ iterations.

**Table 1.** Simulation parameters.

| Parameter | Value |
|---|---|
| RSU coverage area | 2000 m |
| Number of priority tasks | 3 |
| Number of offloaded each priority task | 30 |
| Vehicle speed (m/s) | 20~40 |
| Data rate for vehicle to fog node | 8 Mbps |
| Data rate for vehicle to cloud | 2 Mbps |
| Emergency tasks size (kB) | 5~20 |
| Traffic management tasks size (kB) | 12~30 |
| Infotainment tasks size (kB) | 25~60 |
| Fog node processing capacity (kB) | 500 |

### 5.2. Results and Discussion

In this subsection, the performance of our proposed $ETE_{FN}$ is analyzed and compared with STF, LTF and task processing with FCFS. The comparative results are obtained to calculate the task processing time and number of tasks performed along with their percentages for all three varieties of tasks individually.

Fog-computing nodes are preferred over cloud-computing servers due to the close proximity of vehicles. Fog nodes provide better downloading data rates compared to cloud servers that are remotely placed and require Internet bandwidth to access. The tasks offloaded by vehicles are preferred to be executed by a fog node compared to the remotely

placed cloud servers. Figure 2 shows a comparative analysis of task execution time between cloud servers and fog nodes. It is evident from the results that fog nodes compute the same tasks with considerably less time compared to the cloud servers.

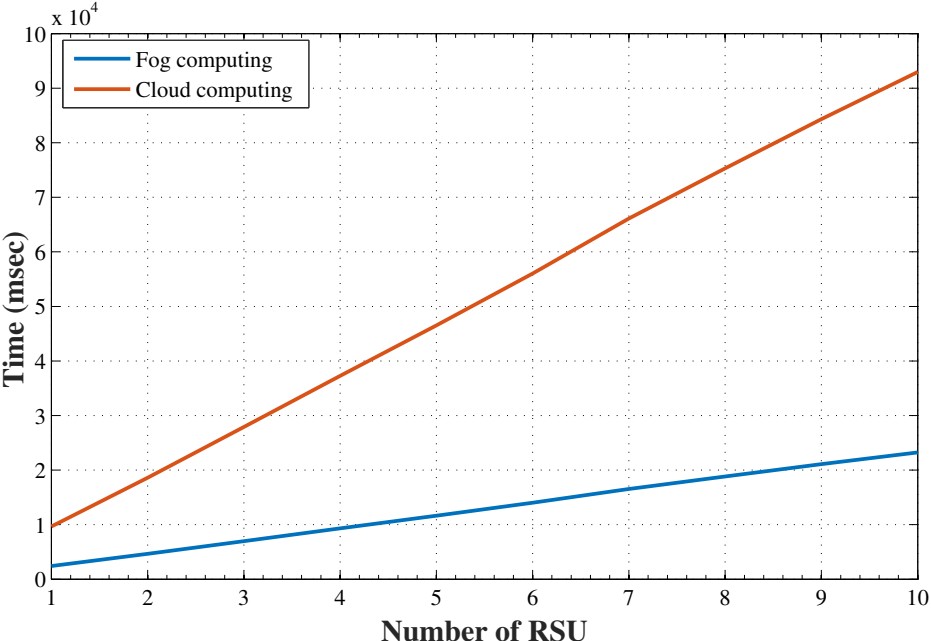

**Figure 2.** Task execution time in fog and cloud servers for varying number of RSUs.

Figures 3 and 4 show a comparative analysis of the number and percentages of tasks performed by fog-computing nodes for a fixed number of offloaded tasks in different processing cycles, respectively. There are 90 offloaded tasks in each processing cycle, 30 each for high, medium and low-priority tasks. Both figures are comprised of three subfigures for three types of offloaded tasks.

Figure 3 shows that for high-priority emergency tasks, $ETE_{FN}$ allows fog computing nodes to compute a maximum number of tasks compared to the other three schemes. For moderate- and low-priority tasks, $ETE_{FN}$ forwards almost all the tasks to the cloud for processing as it almost reaches the processing capacity of fog nodes. However, LTF has the lowest number of high-priority tasks among all schemes. This is due to the fact that the size of most of the low-priority tasks is larger compared to the other two priority types. On the other hand, STF also processes the majority of the high-priority tasks due to their smaller task sizes but less than the proposed scheme. However, FCFS allows the fog node to execute all three types of tasks according to their random arrival.

The results shown in Figure 4 represent the task processing percentages of these schemes for different priorities of tasks individually. The results show that $ETE_{FN}$ executes about 84% of the high-priority tasks in all execution cycles. However, it only execute 3.5% of medium-priority tasks and none of the low-priority tasks is executed, whereas STF executes 70% of the high-priority tasks that is followed by FCFS and LTF. FCFS allows the fog node to process all three types of tasks, and LTF is top in processing most of the low-priority tasks, and executing 38% of these tasks.

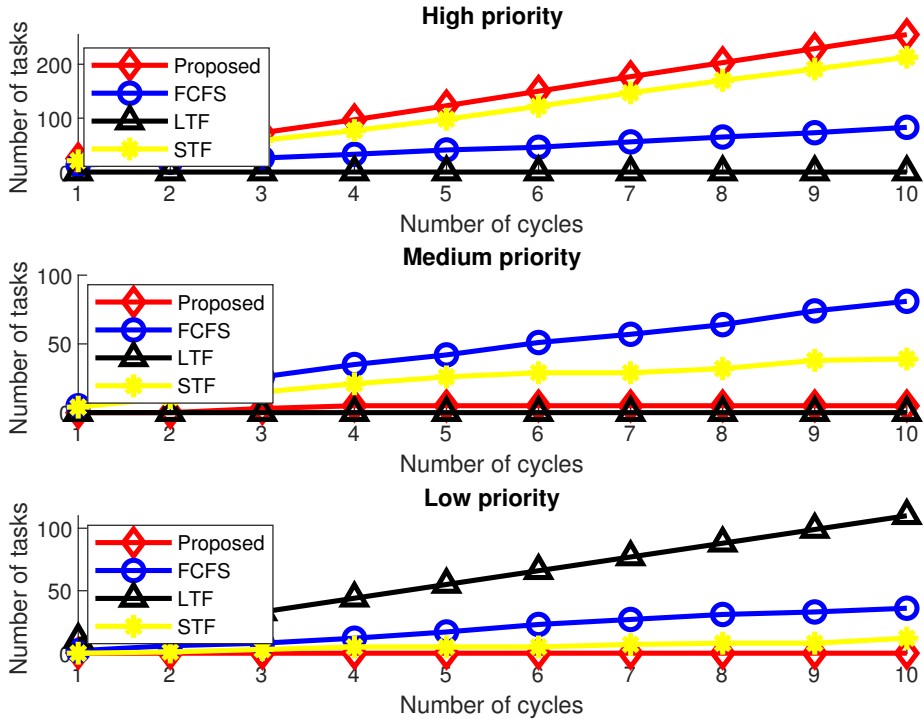

**Figure 3.** Number of tasks performed by fog nodes in different processing cycles.

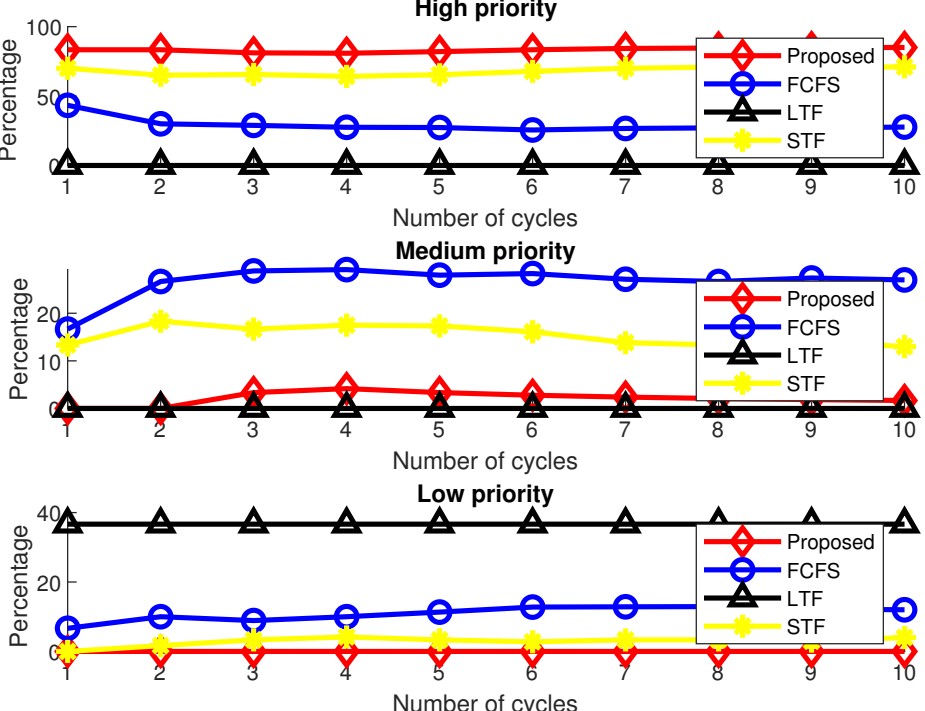

**Figure 4.** Percentage of tasks performed by fog nodes in different processing cycles.

Figures 5 and 6 show comparative performance analysis of $ETE_{FN}$ with the other three schemes for a varying number of offloaded tasks and percentage of tasks performed by fog node in different processing times, respectively. All three different priorities of tasks are shown in subfigures separately. The number of tasks in each figure is incremented by five, and the processed tasks are accumulated by adding the current tasks to previously processed tasks.

Figure 5 shows the number of accumulated tasks processed by fog nodes. The results show that the proposed knapsack-based $ETE_{FN}$ assists fog nodes process all high-priority tasks until the number of tasks reach 25 out of 234 of the 325 processed high-priority tasks at which point the number of high-priority tasks is gradually increased to 55. On the other hand, all three schemes execute a lower number of high-priority tasks. For medium-priority tasks, $ETE_{FN}$ helps the fog node process more tasks compared to the other three schemes in each processing cycle when the number of offloaded tasks is within the range of 20. However, processing tasks in the proposed scheme are less than the other three schemes when number of tasks increases from this limit. This is due to the fact that $ETE_{FN}$ intelligently scrutinizes the tasks for task execution. For low-priority task results, the number of offloaded tasks processed by $ETE_{FN}$ is far less than the other three schemes because $ETE_{FN}$ does not process any of the low-priority tasks because it has already processed most of the high-priority tasks by applying the 0/1 knapsack algorithm.

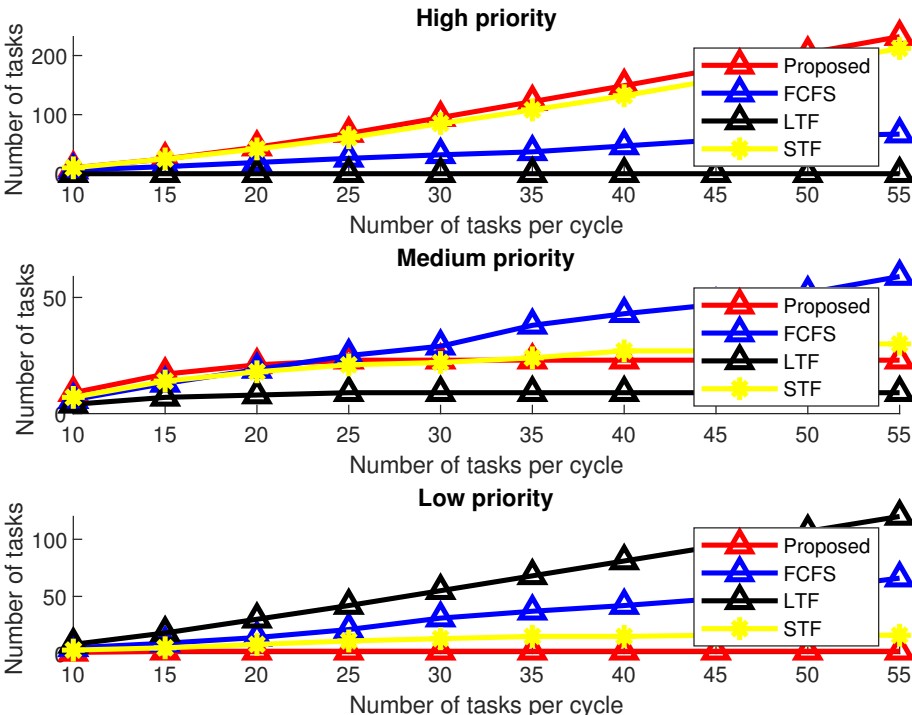

**Figure 5.** Number of tasks performed by fog nodes for varying number of tasks.

Results shown in Figure 6 verify that the fog node performs 100% of offloaded high-priority tasks in $ETE_{FN}$ when the number of tasks is 25 compared to 91%, 38% and 0% of tasks for STF, FCFS and LTF, respectively. When the number of tasks is increased to 55 with an increase of 5 tasks in each cycle, an overall 72% of high-priority tasks are processed compared to 66%, 21% and 0% of high-priority tasks for STF, FCFS and LTF, respectively. For medium-priority tasks, for the initial 10 tasks, the proposed scheme is 100%, more than the rest of the other three schemes. However, for low-priority tasks, the number of processed tasks in $ETE_{FN}$ is less than all three schemes and the processed tasks are at 10% compared to 20%, 70% and 90% of tasks in STF, FCFS and LTF, respectively.

Figure 7 analyzes the task completion time of $ETE_{FN}$ compared to the other three schemes. The total task processing time is defined as the total time required to download the processed task by vehicles from the fog node or from the cloud server. The total processing time is the sum of all processed tasks downloading time of vehicles by adding the previous processing time to current processing time. The results show that the task completion time for high-priority tasks in $ETE_{FN}$ is much less than in STF, FCFS3 and LJF because the majority of these tasks are processed by the fog node itself, and only a few are

forwarded to the cloud. However, accumulated task execution time for low-priority tasks is higher than the rest of the scheme because most of these tasks are forwarded to the cloud for task processing.

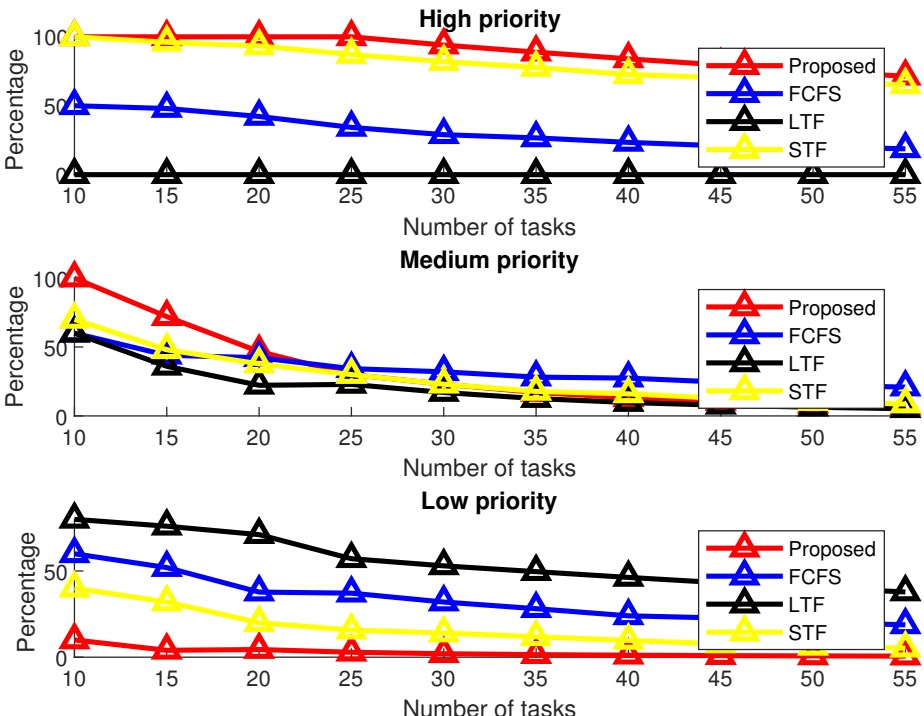

**Figure 6.** Percentage of tasks performed by fog nodes for varying number of tasks.

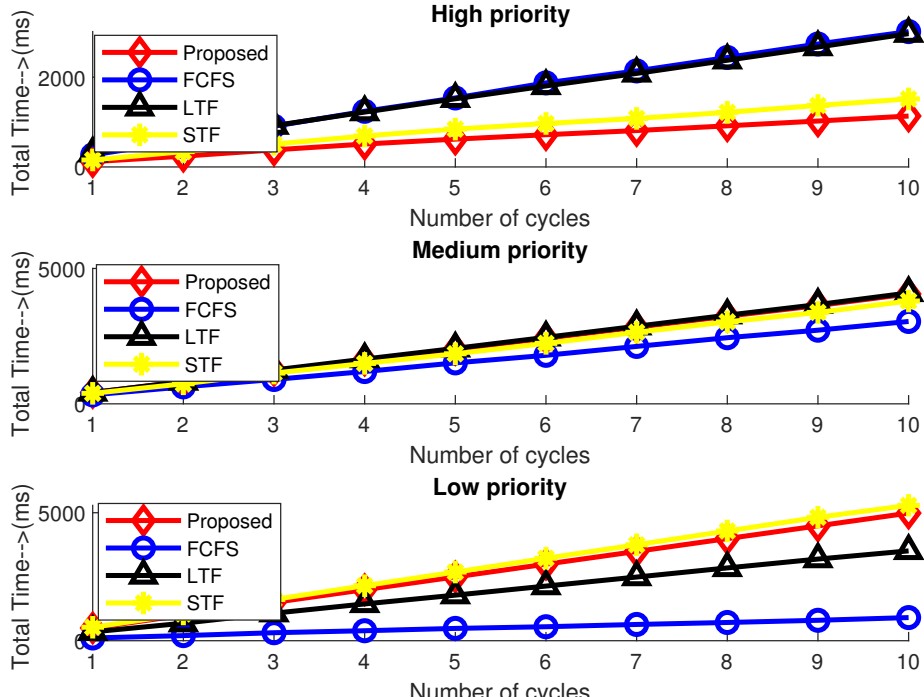

**Figure 7.** Accumulated processing task time for different processing cycles.

It is evident from the results that our proposed scheme accommodates most of the highest priority tasks as compared to all other competitive schemes for varying numbers of processing cycles and for varying tasks in each processing cycle. This results in reduced

execution time for the highest priority offloaded tasks as it allows the fog node to execute most of these tasks within its processing capacity.

## 6. Conclusions

Fog computing nodes are preferred over cloud computing servers for faster processing of tasks. In this paper, we proposed an efficient task execution for a fog node ($ETE_{FN}$) that allowed fog computing nodes to optimally scrutinize the offloaded tasks by applying the knapsack optimization technique. A utility function is calculated for all offloaded vehicular task nodes that are based on their task priority and the time of attachment with the fog node. The key idea is to optimally execute most of the time-sensitive tasks by the fog computing node for faster processing and less sensitive tasks to be forwarded to the cloud. The proposed scheme is compared with FCFS and random processing techniques. The results show that the proposed scheme allows fog nodes to execute more than 98% of high-priority tasks, 47% of medium-priority tasks with an average task processing time of less than 12 ms and 62 ms, respectively, by compromising low-priority tasks.

**Author Contributions:** This article was prepared through the collective efforts of all the authors. Conceptualization, A.N.A., M.A.J., M.H.A.H., M.B.K., A.K.J.S., M.A. and U.F.; critical review, A.N.A., M.A.J., M.H.A.H., M.B.K., A.K.J.S., M.A. and U.F.; writing—original draft, A.N.A. and M.A.J.; writing—review and editing, M.H.A.H., M.B.K., A.K.J.S., M.A. and U.F. All authors have read and agreed to the published version of the manuscript.

**Funding:** The authors extend their appreciation to the Deanship of Scientific Research at Imam Mohammad Ibn Saud Islamic University for funding this work through Research Group no. RG-21-07-06.

**Acknowledgments:** The authors extend their appreciation to the Deanship of Scientific Research at Imam Mohammad Ibn Saud Islamic University for funding this work through Research Group no. RG-21-07-06.

**Data Availability Statement:** The data presented in this study are available on request from the corresponding author.

**Conflicts of Interest:** The authors declare no conflict of interest.

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
