# Peer review of "Intelligent Task Offloading in Fog Computing Based Vehicular Networks"

_applsci, doi:10.3390/app12094521_

Round 1

Reviewer 1 Report

The paper proposes a mechanism to schedule offloaded tasks from vehicles to fog or cloud processing servers. The topic and the proposed technique seem interesting . However, authors should address the following comments:

- Section 2 describes related works, but it is not clear the impact of those works on your proposal. How the works presented in section 2 compare with your proposal, or why do you believe that a comparison is not needed? Related works are also offloading techniques, so a comparison seems relevant, why your work is a better technique than those described in the related work section?

- The performance evaluation is weak. There are numbers for data rates, RSU coverage area, and task size, but where these numbers come from? How do they relate with the 5G network that you claim to use? Additionally, the proposal is compared with two very basic algorithms, so obtaining a better performance is not really a great achievement.   

Author Response

Response to reviewer comments is attached.

Reviewer 2 Report

After carefully readying and analysis of the paper, my comments as as follows:
1. The numerical method used in this paper should be investigated further using more statistical techniques.
2. I didn't find in-depth discussion around the computational cost of the proposed method.
3. The results should be discussed further at the end of the results sections to show the reasoning of the authors.
4. In terms of case-study, I wonder if authors can provide justifications on the suitability to stress test the proposed method.
5. There was no clear justification given on how the proposed method is intelligent?
6. The results should be further explained to demonstrate that accurately proof/disproof the hypothesis.
7. Please highlight the hypothesis of this work in the text.
8. Authors have overlooked many of the recent related works that need to be discussed to justify the need for further research for better method, such as Nonlinear-based chaotic harris hawks optimizer: algorithm and internet of vehicles application; Millimeter-Wave communication for internet of vehicles: Status, challenges and perspectives; Millimeter-Wave communication for internet of vehicles: Status, challenges and perspectives; Smart handoff technique for internet of vehicles communication using dynamic edge-backup node; Transmission power adaption scheme for improving IoV awareness exploiting: evaluation weighted matrix based on piggybacked information.

Author Response

(The authors gave the same response as above.)

Reviewer 3 Report

  1. The article omitted line numbering (for example, 100 - 101).
  2. On page 3, formulas (2) and (3) are given, the values of which are not used in the future. Why did the authors specify these formulas?
  3. In the article in Table 1, parameters that cannot be physically implemented are indicated, for example, Vehicle speed 200 (m/s) is 720 km/h, vehicles do not drive at such a speed. It is necessary to clarify these parameters.

Author Response

(The authors gave the same response as above.)

Reviewer 4 Report

The authors propose an efficient task execution for fog nodes that allowed fog computing nodes to optimally scrutinize the offloaded tasks by applying the knapsack optimization technique. Figure 1 needs to be updated to remove the non-covered area between the two RSUs; Moreover the authors need to explain how they obtained Figure 2. The Abstract section also needs to be rewritten to point out the main findings in the paper. Last but not least, the authors need to explain why their solution is different and better than the solutions already proposed in this field.

Author Response

(The authors gave the same response as above.)

Round 2

Reviewer 1 Report

The authors have partially addressed my previous comments. I still believe that the evaluation part could be significantly better (the proposal is compared with very simple algorithms, and the numbers used to characterize the network are somehow arbitrary).

In the second paragraph of section 5.1 there is a sentence that reads: "observed as The data rate used between vehicle and RSU link is 100 MBytes/s.". This is grammatically wrong and it is not clear how this number is used in the evaluation. Please, clarify/correct this.

Author Response

response attached as pdf.

Reviewer 4 Report

The authors carried out the modifications.

Author Response

response attached as pdf.
